# Characterizing the Motility of Chemotherapeutics-Treated Acute Lymphoblastic Leukemia Cells by Time-Lapse Imaging

**DOI:** 10.3390/cells9061470

**Published:** 2020-06-16

**Authors:** Hsiao-Chuan Liu, Eun Ji Gang, Hye Na Kim, Yongsheng Ruan, Heather Ogana, Zesheng Wan, Halvard Bönig, K. Kirk Shung, Yong-Mi Kim

**Affiliations:** 1Department of Radiology, Mayo Clinic, Rochester, MN 55905, USA; liu.hsiao-chuan@mayo.edu; 2Department of Pediatrics, Division of Hematology, Oncology, Blood and Marrow Transplantation, Children’s Hospital Los Angeles, University of Southern California, Los Angeles, CA 90027, USA; ejiang@chla.usc.edu (E.J.G.); hyekim@chla.usc.edu (H.N.K.); yruan@chla.usc.edu (Y.R.); ogana@usc.edu (H.O.); zwan@chla.usc.edu (Z.W.); 3Goethe University School of Medicine, Institute for Transfusion Medicine and Immunohematology and German Red Cross Blood Service BaWuHe, 60528 Frankfurt, Germany; hbonig@uw.edu; 4Division of Hematology, Department of Medicine, University of Washington School of Medicine, Seattle, WA 98198, USA; 5Department of Biomedical Engineering, University of Southern California, Los Angeles, CA 90089, USA; kkshung@usc.edu

**Keywords:** acute lymphoblastic leukemia, chemotherapeutics-treated, time-lapse imaging, cell motility, integrins

## Abstract

Drug resistance is an obstacle in the therapy of acute lymphoblastic leukemia (ALL). Whether the physical properties such as the motility of the cells contribute to the survival of ALL cells after drug treatment has recently been of increasing interest, as they could potentially allow the metastasis of solid tumor cells and the migration of leukemia cells. We hypothesized that chemotherapeutic treatment may alter these physical cellular properties. To investigate the motility of chemotherapeutics-treated B-cell ALL (B-ALL) cells, patient-derived B-ALL cells were treated with chemotherapy for 7 days and left for 12 h without chemotherapeutic treatment. Two parameters of motility were studied, velocity and migration distance, using a time-lapse imaging system. The study revealed that compared to non-chemotherapeutically treated B-ALL cells, B-ALL cells that survived chemotherapy treatment after 7 days showed reduced motility. We had previously shown that Tysabri and P5G10, antibodies against the adhesion molecules integrins α4 and α6, respectively, may overcome drug resistance mediated through leukemia cell adhesion to bone marrow stromal cells. Therefore, we tested the effect of integrin α4 or α6 blockade on the motility of chemotherapeutics-treated ALL cells. Only integrin α4 blockade decreased the motility and velocity of two chemotherapeutics-treated ALL cell lines. Interestingly, integrin α6 blockade did not affect the velocity of chemoresistant ALL cells. This study explores the physical properties of the movements of chemoresistant B-ALL cells and highlights a potential link to integrins. Further studies to investigate the underlying mechanism are warranted.

## 1. Introduction

Patterns of motility in human leukemia cells based on time-lapse cinematography were studied as early as 1978 [1]. Three basic modes of cell motility were evaluated: surface motility, spot motility and locomotion. The relationship between the different patterns of motility and the clinical course of the disease still remains to be investigated. The tumor microenvironment may provide selective pressure for pre-existing mutants within the population [2,3,4]. An additional contribution to accelerating the evolution of cancers is the mutagenic stress response followed by the emergence of adaptive phenotypes [5,6]. In particular, the resistance of metastatic cancer cells to chemotherapy is a driver of mortality in cancer [7]. This resistance is correlated with the motility of the mutant cells in a chemotherapy gradient with the selective pressure of mutagenic chemotherapy. It was also demonstrated that MDA-MB-231 cancer cells treated with 200 nM doxorubicin move toward higher drug concentrations, followed by proliferation [7]. Mutagenic drug gradients in the microenvironment could lead to the accelerated evolution of drug resistance if the cells are motile across the gradient [6,8]. Therefore, it is crucial to explore motility in leukemia cells that are treated with chemotherapeutic drugs.

The bone marrow (BM) microenvironment regulates the growth of hematopoietic stem cells (HSCs) and malignant cells by direct physical contact, acting as a “shelter” and releasing factors through various signaling pathways [9,10,11,12,13,14,15]. The interaction between mesenchymal stromal cells (MSCs) and leukemia cells has been of much interest. Leukemia cells have been described to remodulate the BM microenvironment via stimulating MSCs to release more ActivinA, a pleiotropic cytokine that belongs to the TGF-β superfamily [16]. ActivinA, in turn, caused enhanced cell motility, migration toward CXCL12, and the invasion of B-cell precursor (BCP)-ALL cells as determined by time-lapse microscopy. Furthermore, this in vitro finding was verified by the enhanced in vivo engraftment of BCP-ALL cells to BM and extramedullary sites (peripheral blood, spleen, liver, meninges, and brain) in a xenograft mouse model [16]. Recent developments in imaging technologies have provided powerful new tools to visualize, image, track, and analyze HSCs and their interactions with the BM niche by labeling with fluorescent dyes such as Qdot 655 [17].

Integrins, a form of cell adhesion receptors, are highly expressed in leukemia cells, including integrin α4 and α6 [18,19]. Our previous preclinical studies demonstrated the successful eradication of primary B-cell acute lymphoblastic leukemia (ALL) cells engrafted in NSG (NOD. Cg-*Prkdc^scid^ Il2rg^tm1Wjl^*/SzJ) mice by treatment with chemotherapeutic drugs combined with integrin blockade [18,19]. To characterize the physical properties of the chemotherapeutics-treated B-ALL cells, we used acoustic tweezers to measure the deformable capacity, which was found to be increased by the chemotherapeutic drug combination [18]. Furthermore, compared with its isotype control (IgG1), the antibody against α6 (P5G10) but not Tysabri (humanized antibody against α4) markedly reduced the deformability [19]. However, the role of the motility of chemotherapeutics-treated ALL cells has not yet been widely explored. In the present study, the primary B-ALL cells, LAX7R, LAX56, and ICN24, were monitored using time-lapse cinematography for 12 h to investigate their motility in vitro.

## 2. Materials and Methods

### 2.1. System Structure of Time-Lapse Cinematography

The time-lapse system consists of four sections, including the gas/air route, temperature feedback route, temperature control route, and light route, which are marked in blue, brown, green, and red, respectively, in Figure 1. For the gas/air route, CO_2_ was injected into a CO_2_ controller with an air heating function (CTI-controller, 3700 digital, PeCon GmbH, Erbach, Germany) to generate 37.5 degree 5% CO_2_ gas, which then entered a heatable universal mounting frame to establish a suitable environment for the cells. Additionally, air was heated to 37 °C using a heating unit (PeCon GmbH, Erbach, Germany) and injected into a large chamber covering the entire CO_2_ incubation system, as indicated by the blue arrow in Figure 1. A temperature controller (Tempcontrol 37-2 Digital, PeCon GmbH, Germany) with two independent channels was used to manage the heating unit and the CTI-3700 CO_2_ controller and to provide the temperatures required for the experiments, as represented by the green arrow in Figure 1. The temperature feedback route is shown by the brown arrows. Two temperature sensors were placed in the universal mounting frame and the inside wall of the large chamber to send real-time temperature information back to the heating unit and CTI-3700 CO_2_ controller to balance the temperature in the system.

For the light route, shown by the red arrows (Figure 1), a fluorescence illuminator equipped with a 120 W lamp (EXFO X-Cite120, Excelitas Technologies, Inc., Covina, CA, USA) was used to apply uniform wide-field fluorescence illumination to a Zeiss Axiovert 200 inverted fluorescence phase contrast microscope. Two fluorescent proteins, green fluorescent protein (GFP) and mCherry fluorescent protein, were utilized in the study. A 475 nm (blue) light was used to excite GFP and generate an emission peak at 509 nm (green) for monitoring ALL cells, and a 587 nm (red) light was used to excite mCherry to obtain an emission spectrum at 610 nm for observing stromal cells. A fluorescein isothiocyanate (FITC) filter was applied to separate the two spectra. Twelve-bit gray level images with a resolution of 1344 × 1024 were captured with a digital camera (C4742-80-12AG ORCA-ER, Hamamatsu, Shizuoka Prefecture, Japan) controlled by a camera controller (C4742-80-12AG ORCA-ER, Hamamatsu, Japan) and finally stored on a computer. All monitoring parameters for the primary ALL cells were assessed via the user interface of the Micro Manager Software version 1.4.15 (San Francisco, CA, USA).

### 2.2. Patient Bone Marrow Samples, Cell Preparation, and Cell Monitoring Procedure

Bone marrow (BM) samples from B-ALL patients were obtained in compliance with the Institutional Review Board regulations of each institution. Informed consent was obtained from all human subjects. Leukemia cells were processed and cultured as previously described [19]. To characterize the motility of chemotherapeutics-treated ALL cells, three primary B-ALL cells—LAX7R, LAX56, and ICN24—which were derived through xenografting patient B-ALL cells in NSG mice, were transduced with lentiviral pCL6 IRES GFP (GFP+) and cocultured at a density of 5 × 10^4^/cm^2^ with or without irradiated (30 Gy) mCherry-HS27a (4 × 10^4^/cm^2^) (human stromal cells). Chemotherapeutics-treated B-ALL cells were defined as those that survived after chemotherapy treatment. Cells were treated with VDL (vincristine 5 nM, dexamethasone 0.05 nM, and l-asparaginase 0.0025 IU) (treatment group) or left without treatment (medium control group) for 7 days. Furthermore, two antibodies—P5G10 (a gift from Dr. Elizabeth Wayner) and natalizumab (Tysabri, Elan Pharmaceuticals/Biogen Idec, San Diego, CA, USA), both at a concentration of 20 µg/mL—were used to explore the motility of the three types of primary ALL cells after chemotherapy treatment and in combination with the blockade of integrin α6 or α4, respectively.

All leukemia cells were washed two times with DPBS and plated in medium without any drugs into 8-well chamber slides (ibidi, Martinsried, Planegg, Germany) for the time-lapse imaging assay for 12 h. The cells were arranged in the 8-well chamber as shown in the lower right corner of Figure 1. The chamber was placed on the heatable universal mounting frame at 37.5 degrees centigrade in a 5% CO_2_ environment for monitoring. To track the motility of the cells, z-stack scanning was performed with a 20 × 0.8 objective lens for 12 h every 15 min. Each layer in each well was exposed to 475 nm (blue) and 587 nm (red) light at the same time.

### 2.3. Cell Viability Measurement

The viability of the ALL cells after VDL, Tysabri, or P5G10 treatment was determined by Annexin V-PE (BioLegend, San Diego, CA, USA) and 7-aminoactinomycin D (7-AAD) (BioLegend) staining and flow cytometry (FACScalibur, BD Biosciences, San Diego, CA, USA). The FlowJo 7.6.5 software (FlowJo, LLC, Franklin Lakes, NJ, USA was used to analyze the acquired data.

### 2.4. Imaging Processing and Reconstruction of Time-Lapse Cinematography

The z-stack images were visualized and processed using Fiji ImageJ version 1.51f (National Institutes of Health, Bethesda, MA, USA). All raw data were imported to a bulk stack image and then transferred to a hyperstack image (composite mode) with two-channel (green and red color) and serial *z*-axis information for tracking cells. Time-lapse imaging included all GFP+ ALL cells. However, membrane-intact rounded, live ALL cells were counted for motility analysis. Cell tracking was performed using MTrackJ, a plugin tool for Fiji ImageJ. After tracking, the velocity and total migration distance of the cells were calculated. Meanwhile, to obtain a better image quality, a deconvolution algorithm was performed by utilizing the AutoQuant X3 software (Media Cybernetics, Inc., Rockville, MD 20852, USA) to deblur the images. A 3D projection algorithm was used to convert serial bulk stack images including *z*-axis information to a 3D rotational image to observe the motility of the cells from a side view.

### 2.5. Statistical Analyses

Unpaired *t*-tests were used to determine the differences between two groups. One-way ANOVA followed by post hoc analysis (Tukey’s test) was applied to analyze the differences among 3 or more groups in GraphPad Prism 5 (GraphPad Software Inc., San Diego, CA 92108, USA). * *p* < 0.05 was defined as a significant difference.

## 3. Results

### 3.1. The Motility of Primary Pre-B ALL Cells versus Chemotherapeutics-Treated ALL Cells Based on Time-Lapse Cinematography

The motility of the three primary groups of B-ALL cells, including LAX7R, LAX56, and ICN24, was characterized. Two of the cell groups (LAX7R and LAX56) were obtained upon relapse after chemotherapy, and the remaining cells (ICN24) were obtained at the time of diagnosis. The status and cytogenetics of the ALL are shown in Table 1. Each type of cell was separated into two conditions: leukemia cells in medium (vehicle control) and in VDL (chemotherapy treatment). Of note, as the stromal cells are irradiated to prevent cell division and crowding of the tissue plate, chemotherapy in the dose applied did not have cytotoxic effects on them. Each condition was then divided into two groups: leukemia cells only and leukemia cells plated onto HS27a human stromal cells to investigate the motility of B-ALL cells with or without stromal support under chemotherapeutics-treated conditions. Figure 2a–d depict representative images that demonstrate the velocity and migration distance of LAX7R cells plated with HS27a human stromal cells in medium. It should be noted that the mCherry HS27a cells are not present in the images to illustrate the motility of the ALL cells. The red lines in both images represent the tracked migration path of a single cell. The results show that their trajectory seems to be random and that the cells can move anywhere in the chamber.

By contrast, ALL cells treated with VDL for 7 days exhibited lower motility, as illustrated in Figure 2b for LAX7R. Their trajectories seem to be limited to either remaining at the initial cell location or migrating to a new location first and then remaining there. We also investigated how leukemia cells migrate both in the medium control and VDL groups without stromal cells. The viability of the primary B-ALL cell (LAX7R) was determined prior to imaging, showing that compared to the control group, the VDL-treated group had a significantly reduced cell viability; nevertheless, some viable cells remained and were imaged (Figure 2d).

A vector plot was used to illustrate the motility tracks of the leukemia cells (Figure 2c) and visualize the motility of both control and VDL-treated ALL cells simultaneously. The numbers on the circumference of the largest circle represent the z-stack images, or steps, and the numbers on the concentric circles represent how far the leukemia cells moved for each step. The numbers on the outside circle represent indices of the individual z-stack images, or steps, for a total of 48 steps. The numbers on the concentric circles represent the distance that a cell can travel from its origin, from 0 to 90 µM. The plot demonstrates that the migration distance of the cells treated with VDL (red line) is smaller than that of the cells in the control medium (blue line), which suggests that the VDL-treated cells exhibited short migration distances and stayed close to their origins.

To further characterize the motility of the VDL-resistant B-ALL cells, two features were analyzed: cell velocity and migration distance from the origin. Figure 3a,c,e (left panels) represent the velocities of the three B-ALL cells (LAX7R, LAX56, and ICN24) cocultured with HS27a human stromal cells (+HS27a) and demonstrate that the chemoresistant cells were slower than the medium control cells (*p* < 0.001 for all types).

Furthermore, the results also revealed a smaller migration distance from the origins for all three leukemia cells cocultured with human stromal cells than for the medium control group cells (*p* < 0.0001 for all types) (Figure 3b,d,f left panels, +HS27a). For the case of the ALL cells alone (without co-culture with human stroma cells, -HS27a), all types of ALL cells in medium exhibited a greater velocity than those in VDL (Figure 3a,c,e right panels); however, the migration patterns were not significantly different in LAX7R and LAX56 cells but only in ICN24 cells (Figure 3b,d,f -HS27a). Taken together, ALL cells without stromal cell co-culture displayed greater velocities than ALL cells cocultured with HS27a cells regardless of their plating in medium or VDL (*p* < 0.0001 for all cases), which means that co-culturing with stromal cells impacts the motility of primary ALL cells.

### 3.2. The Effect of Antibodies Against Integrins α4 and α6 on Chemotherapy-Treated, Viable Primary B-ALL Cells

Since we showed that stromal cells impact the motility of primary ALL cells, we next sought to determine the potential effect of de-adhering antibodies on this motility. Two antibodies, P5G10 (against integrin α6) [18] and Tysabri (against integrin α4) [19], were used to investigate the effects of integrin blockade on the motility of chemotherapeutics-treated ALL cells (LAX56). Three experiments were performed, and the pooled data are shown in Figure 4. Figure 4a,d,g show the viability of three B-ALL cells treated with VDL combined with either of the two antibodies (P5G10 or Tysabri). P5G10 itself but not Tysabri may moderately decrease the viability of ALL cells (Figure 4a), which has been described previously after 2–4 days of treatment [20,21]. Significant changes in velocity were observed between cells treated with either VDL alone and those treated with VDL in combination with Tysabri in two of the three cases (LAX7R and LAX56) (Figure 4b,e). In the same cases, the Tysabri + VDL treatment resulted in a significantly decreased migratory distance for LAX7R (Figure 4c) and LAX56 (Figure 4f) (*p* < 0.001 for VDL vs. Tysabri + VDL). P5G10 in combination with VDL did not affect velocity in all three cases (Figure 4b,e,h) compared to VDL only but decreased migratory distance in two cases—LAX7R (Figure 4c) and ICN24 (Figure 4i)—but not for LAX56 (Figure 4f).

## 4. Discussion and Conclusions

### 4.1. Chemotherapy-Treated ALL Cells Showed Decreased Motility

The present study indicates that drug treatment in ALL cells is inversely correlated with the mechanical properties of the cells: chemotherapeutics-treated ALL cells exhibited a reduced velocity and shorter migration distance than untreated cells (Figure 2). We defined chemotherapeutics-treated B-ALL cells as those that survived chemotherapeutic treatment. Although the ICN24 B-ALL cells were taken at the time of disease diagnosis (Table 1), we are unsure if the patient eventually relapsed or if they were responsive to chemotherapy long-term. Follow-up studies should ideally include both diagnosis and relapse samples to determine any differences in motility.

VDL, the chemotherapeutic agents used here, were initially thought to affect cell motility and were selected for our studies because they form part of the major backbone of the induction block by ALL treatment regimens. Vincristine, in particular, inhibits microtubule formation in the mitotic spindle, resulting in an arrest of cell division at the metaphase stage. Future studies should be performed with drugs that do not primarily affect motility to validate our findings.

T-ALL cells have shown to directly interact with CXCL12-producing MSCs in a dynamic BM microenvironment, where leukemic cells were highly nonmotile and strongly associated with stromal cells [22]. Hawkins et al. investigated the migratory property of xenotransplanted primary human T cell acute lymphoblastic leukemia (T-ALL) in NOD/SCID/γ mice using time-lapse intravital microscopy [23]. In contrast with our results, they demonstrated the significant enhanced migration velocity of chemoresistant cells with cell division events immediately after the third dose of VDL, suggesting that faster migration and a lack of long-lasting interactions with the BM microenvironment are conserved characteristics of chemoresistance in T-ALL [23]. However, this observation was made after a 2 day treatment with VDL. This is a critical difference, as it is possible that at this earlier time point, cytotoxic events are still occurring and not all cells are fully chemoresistant. In other words, a mixed cell population of viable drug-resistant cells and dying cells may have been studied instead of a predominantly drug-resistant cell population. By contrast, in our model, 7 days of chemotherapy were chosen, the timepoint where viability stops decreasing and a state of surviving, predominantly drug-resistant cells is achieved. This may be one explanation for the difference to our results. Of note, it was also shown using intravital imaging that AML cells are migratory, and in contrast with T-ALL, chemoresistant AML cells become less motile [24]. Further studies are warranted to distinguish motility differences in different subtypes of leukemia and different drug treatments to further elucidate the effects on motility.

We had previously demonstrated that primary B-ALL cells treated with VDL exhibited increased cell deformation and that 7 days of VDL treatment killed the majority of ALL cells [18]. Considering the results of both the previous and the current study, we may conclude that the drug-treated ALL cells—of which some would get killed under the continuous drug condition whereas a few would become drug-resistant clones—move slowly and with greater elasticity. The possibility of the drug-treated ALL cells containing drug-resistant clones has been demonstrated in our previous study [25], in which in vitro VDL-treated cells relapsed in several ALL cases. One of the reasons for this could be that drug-treated ALL cells become entangled with human stromal cells (HS27a) via direct contact support, providing a survival benefit. Another reason could be that stroma cells, as they shelter leukemia cells from chemotherapy, impact the velocity and migration patterns of ALL cells (Figure 2 and Figure 3). It has been reported that the enhanced motility of leukemia cells prevents the recurrence of disease in the central nervous system (CNS) [26]. The authors indicated that motile cells have the ability to re-enter the bloodstream and become exposed to chemotherapy; by contrast, immotile lymphoblasts remain behind the blood-brain barrier to escape chemotherapy and eventually give rise to a CNS relapse of leukemia. Although it might be possible that CNS relapse originates from ALL cells with higher motility, based on this report, chemotherapeutics-surviving ALL cells, which would contain drug-resistant clones, potentially exhibit a slower velocity and less motility than control cells, implying that these ALL cells may stay in their original location to avoid egress to the bloodstream and exposure to chemotherapy. Thus, chemotherapy-surviving ALL cells with altered mechanical properties may contribute to ALL relapse. This requires follow-up studies.

### 4.2. The Roles of Integrin Blockade in Cell Motility

Integrins play a crucial role in linking the extracellular and intracellular skeletons. Cytoplasmic linker molecules are thought to mediate integrin-actin binding and likely act as potential dynamic mediators of cell anchorage [27]. Given that the organization of actin filament networks is critical for cell motility, we hypothesized that integrin-actin binding plays a role in the motility and deformability of leukemia cells. The blockade of integrins α4 and α6 by their corresponding antibodies (Tysabri and P5G10, respectively) changed the velocity and migratory distance in the same two of the three tested chemoresistant (chemotherapy surviving) ALL cells (Figure 4b,c,e,f). Interestingly, the combination of Tysabri with VDL tended to increase the migratory distance of one of the three VDL-treated ALL cells (ICN24; Figure 4i) compared with that of cells treated with VDL alone and decreased the distance in two out of the three ALL cases (LAX7R and LAX56). P5G10 did not affect velocity in all three cases but decreased the migratory distance in two of the three cases (LAX7R and ICN24). We have shown previously that Tysabri de-adhered leukemia cells from the stromal cells, resulting in the discontinued direct contact and niching of the cells and a tendency to increase migratory distance [21]. The reasons for the inability of P5G10 to cause similar effects are unclear. One difference is that P5G10 but not Tysabri decreases viability significantly at 4 days in in vitro assays [20,21], indicating different roles of the two integrins in the viability and physical properties of ALL cells. This finding corroborates our recent research showing that integrins α4 and α6 have potentially different roles in drug resistance [28]. The link between the mechanical properties and drug resistance of ALL cells will be investigated and discussed in future research.

## Figures and Tables

**Figure 1 cells-09-01470-f001:**
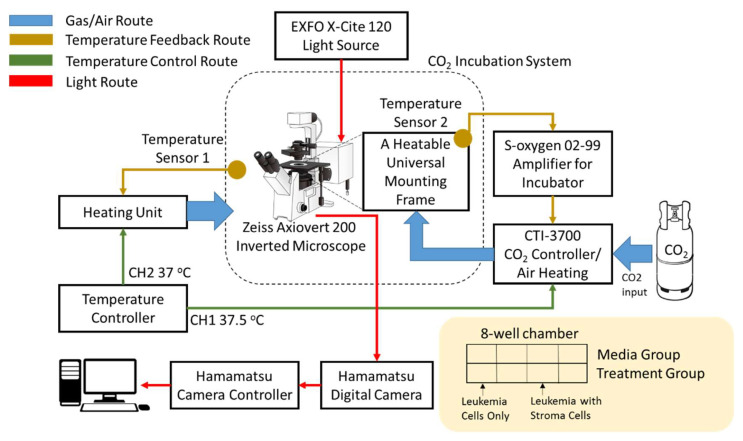
The schematic of the whole system framework for time-lapse cinematography is shown. The system consists of a gas/air route indicated with blue arrows, a temperature feedback route indicated with brown arrows, a temperature control route indicated with green arrows, and a light route indicated with red arrows. An 8-well chamber plated with leukemia cells and human stromal cells was used to monitor the motility of primary ALL cells.

**Figure 2 cells-09-01470-f002:**
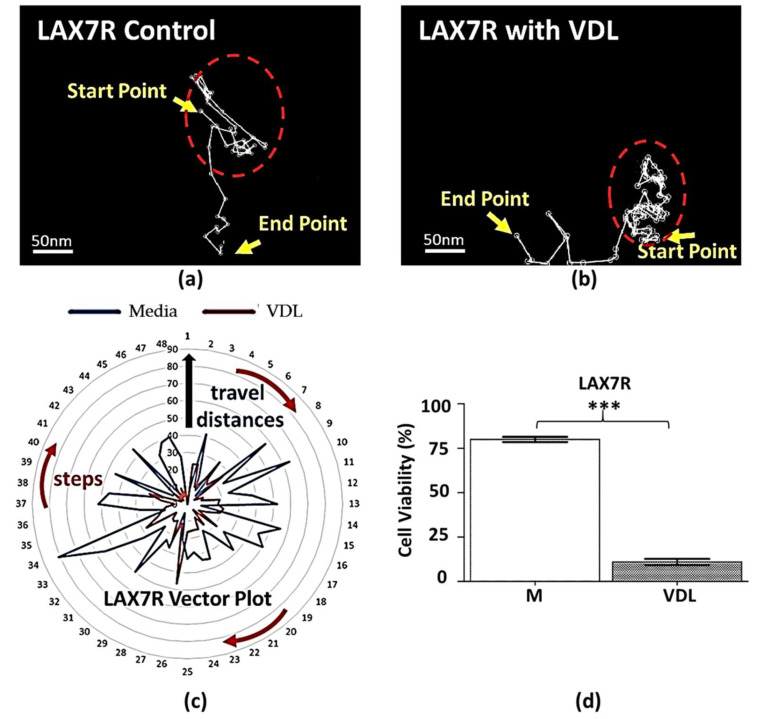
An example of LAX7R co-cultured with HS27a human stromal cells monitored by time-lapse microscopy to display the motility tracks of viable primary B-ALL cells in control medium and treated with chemotherapy. (**a**,**b**) illustrate a case of a LAX7R cell migration pattern (white lines) in medium control and with VDL chemotherapeutical treatment for 7 days. The time-lapse image reveals that the migration pattern is tangled at the start point of the migration and displays a weak motility as the cells were treated with VDL (red-dashed circles). The scale bars in (**a**,**b**) are 50 nm. (**c**) A proposed vector plot provides a visualization to simultaneously observe cell motility and migration patterns in both medium and VDL. The arc (red arrows) and radial (blue arrow) indicate a cell’s migration steps and travel distance from its start point. In the study, the 48 steps (12 h recording) were considered in both groups. The travel distance to 90 indicates 26.1 µm as the actual distance. (**d**) The viability of the medium control and VDL-treated cells on Day 7 was measured by 7-AAD and Annexin V-PE staining using flow cytometry. *** *p* < 0.001 compared with the medium group, unpaired *t*-test. The letter M stands for medium.

**Figure 3 cells-09-01470-f003:**
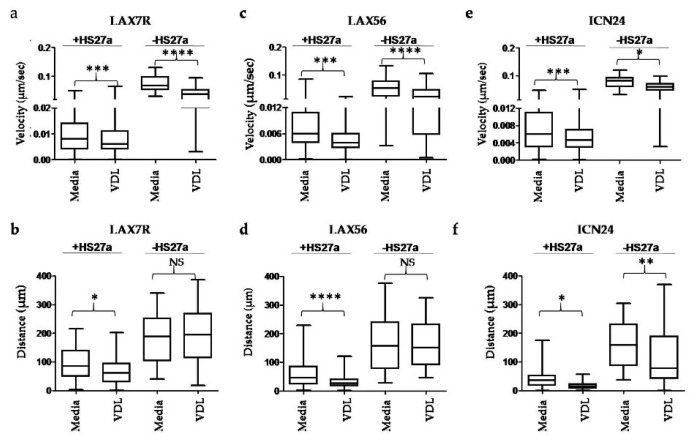
Effect of chemotherapeutic treatment of primary ALL cells cocultured with human stromal cells on velocity and migratory distance. Velocities of (**a**) LAX7R, (**c**) LAX56, and (**e**) ICN24 cells treated with medium or VDL. Cells were co-cultured with HS27a human stromal cells (+HS27a) or not (-HS27a). The migration distance from the origins of the (**b**) LAX7R, (**d**) LAX56, and (**f**) ICN24 cells that were or were not cocultured with HS27a cells was also measured. * *p* < 0.05, ** *p* < 0.01, *** *p* < 0.001, and ***** p* < 0.0001 when compared with their corresponding medium controls (unpaired *t*-test). NS, not significant.

**Figure 4 cells-09-01470-f004:**
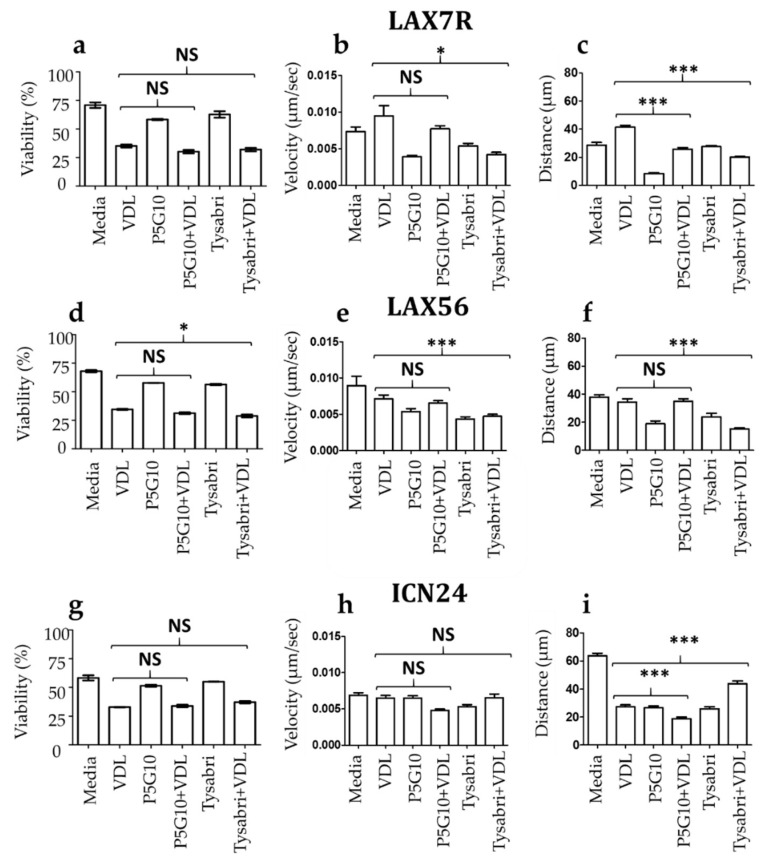
The effects of integrin blockade on the velocity and migration distance of primary B-ALL cells. B-ALL cells LAX7R (**a**–**c**), LAX56 (**d**–**f**), and ICN24 (**g**–**i**) were treated with VDL (VDL) or without VDL (Medium) along with P5G10 or Tysabri for 7 days. After a washing and medium incubation period of 12 h, ALL cells were time-lapse imaged for 12 h (**a**,**d**,**g**). Cell viability was measured by 7-AAD/Annexin V staining using flow cytometry (**b**,**e**,**h**). The velocities and (**c**,**f**,**i**) migratory distances of the three ALL cells are depicted. The differences were analyzed by one-way ANOVA with Tukey’s test. * *p* < 0.05 and *** *p* < 0.001. NS, not significant. The pooled data of three experiments are shown.

**Table 1 cells-09-01470-t001:** Status and cytogenetics of the B-ALL cells.

ALL	Status	Cytogenetics
LAX7R	Relapse	KRAS^G12V^
LAX56	Relapse	*t*(Y;7) (p1.3; p13)
ICN24	Diagnosis	Normal

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
