# Peer review of "Characterizing the Motility of Chemotherapeutics-Treated Acute Lymphoblastic Leukemia Cells by Time-Lapse Imaging"

_cells, 2020, doi:10.3390/cells9061470_

Round 1

Reviewer 1 Report

The effect of the chemotherapy drug is to induce cell proliferation arrest, cell death,  particularly in B-ALL. Also, the dying cells,  of course, have dramatically reduced cell motility. Moreover,  treatment of the patients' cells for 7 days and then observe the cell mobility in 12 hrs, these approaches are very difficult to get reliable results because almost all cells are dead cells or dying cells, not the drug-resistant cells. Therefore, this manuscript is not acceptable to be be published. 

 It is better to re-design the experiments and get reliable results.

Author Response

Point-by-Point Response

cells-794823

05/22/20

Reviewer #1

Comments and Suggestions for Authors

The effect of the chemotherapy drug is to induce cell proliferation arrest, cell death,  particularly in B-ALL. Also, the dying cells,  of course, have dramatically reduced cell motility. Moreover,  treatment of the patients' cells for 7 days and then observe the cell mobility in 12 hrs, these approaches are very difficult to get reliable results because almost all cells are dead cells or dying cells, not the drug-resistant cells. Therefore, this manuscript is not acceptable to be be published. 

 It is better to re-design the experiments and get reliable results.

Response: We thank the reviewer for her/his review and comments.  We  would like to set up experiments to address the  concerns. However, due to limitations in personnel, time and resources especially at this moment in human history, we currently cannot.  To still explain our  model: We treated the B-ALL cells for 7 days with chemotherapeutic drugs (combination of vincristine, dexamethasone and L-asparaginase) followed by measuring their motile property because this treatment could kill majority of the cells and the surviving cells were mainly under the phase of chemoresistance , which we have published previously (Gang EJ et al. Oncogene.2014, PMID: 23728349; Figure 5A as an example). We will follow-up with further investigations distinguishing drug resistant cells from  dying cells.

Reviewer 2 Report

  The revised version was improved.

Author Response

Reviewer #2

Comments and Suggestions for Authors

The revised version was improved.

Response: We thank the reviewer for her/his review and comments.

Reviewer 3 Report

The authors have addressed every comment. Regarding the quality of images, however, this has not been improved, I would say that some of them are displayed even with worse quality than in the previous manuscript. This should be amended before final acceptance.

Author Response

Reviewer #3

Comments and Suggestions for Authors

The authors have addressed every comment. Regarding the quality of images, however, this has not been improved, I would say that some of them are displayed even with worse quality than in the previous manuscript. This should be amended before final acceptance.

Response: We thank the reviewer for her/his review and comments.  We have significantly improved the quality of Figures 2,3 and 4.

Reviewer 4 Report

The authors studied a relevant topic and have appropriate tools to make interesting and original observations. Unfortunately, the reported experiments and results are limited, poorly described and can not support conclusions.

Specific comments:

- The submitted manuscript is annotated which makes the reviewing process harder.

- The submitted images are of very poor quality and not suitable for publication. For exemple, it is not possible to evaluate the cell tracks in Figure 2a-d and Figure 4 is not readable.

- No information is provided regarding how the primary B-ALL cells were obtained from patients, including institutional ethical approval.

- More methodological details are needed, such as the cell seeding density for B-ALL and stromal cells

- The authors report that chemotherapy (“VDL”)-resistant LAX7R cells are less motile in vitro. However, this may be a consequence of decreased cell viability. Cell viability is check by flow cytometry but the imaging is done on all GFP+ cells. Motility measurements may therefore be biased by dead GFP+ cells.

- What was the impact of chemotherapy on stromal cells?

- Are there differences in the type of movement? (stochastic/random or directed)

- Motility measurements may be signigficantly influenced by the number of cells (both leukemia and stromal cells) in each well. Have the authors looked at this?

- What does Figure 2e refers to? Viability with or without stromal support?

- Was the cell viability of LAX56 and ICN24 checked? Also, I would expect much higher viability of ICN24 cells as these are from a relapse sample. Have the authors checked this?

- From the images in Figure 2 it seems that there are more chemoresistant LAX7R cells without stromal support, which is counterintuitive. Please explain.

- The discussion of figure 4 is very confusing.

- What was the rationale to test integrin blockage? Was the experiment in figure 4 done on stromal support? Do stromal cells express the integrin ligands?

- I would expect if integrins are relevant for the adhesion of B-ALL cells that their inhibition resulted in a significant increase in migration, as shown for the abscense of stromal support.

- Previous data that are very relevant for the type of biological phenomena the authors are studying were overlooked (check the papers: PMID 26058075, 27750279, 30262563). I suggest that the authors perform an extensive literature review on the topic and redefine their references.

Author Response

Reviewer #4

Comments and Suggestions for Authors

The authors studied a relevant topic and have appropriate tools to make interesting and original observations. Unfortunately, the reported experiments and results are limited, poorly described and can not support conclusions.

Response: We thank the reviewer for her/his review and comments.  We revised the manuscript addressing the reviewer’s comments below.

Specific comments:

- The submitted manuscript is annotated which makes the reviewing process harder.

- The submitted images are of very poor quality and not suitable for publication. For exemple, it is not possible to evaluate the cell tracks in Figure 2a-d and Figure 4 is not readable.

Response: We agree and revised Figures 2, 3 and 4.

- No information is provided regarding how the primary B-ALL cells were obtained from patients, including institutional ethical approval.

Response: The missing information is included in line 109~112.

- More methodological details are needed, such as the cell seeding density for B-ALL and stromal cells

Response: The missing information is included in line 114~116.

” B-ALL cells in NSG mice, were transduced with lentiviral pCL6 IRES GFP (GFP+) and cocultured at a density of 5x104/cm2 “ and  “mCherry-HS27a (4x104/cm2)“

- The authors report that chemotherapy (“VDL”)-resistant LAX7R cells are less motile in vitro. However, this may be a consequence of decreased cell viability. Cell viability is check by flow cytometry but the imaging is done on all GFP+ cells. Motility measurements may therefore be biased by dead GFP+ cells.

Response: We clarified that live and dead cells are distinguished by cell shape in line142-143:.

“Time-lapse Imaging included all GFP+ ALL cells. However, membrane-intact rounded, live leukemia cells were counted for motility analysis”

- What was the impact of chemotherapy on stromal cells?

Response: As the stromal cells are irradiated to avoid cell division and crowding of the tissue plate, chemotherapy in the dose applied did not have cytotoxic effects on them. This information is now included in the discussion in line 163-165.

- Are there differences in the type of movement? (stochastic/random or directed)

Response: This is an excellent question. Cell migration is a very complex behavior. This type of movement could be stochastic. David B. Brückner et al. report that cells in two-state micropatterns exhibit intricate nonlinear stochastic migratory dynamics (Nature Physics volume 15, pages 595–601, 2019). T. Kwon et al. also report that cell migration should be not only stochastic but also heterogeneous by the calculation using stochastic differential equation (Scientific Reports volume 9, Article number: 16297, 2019).  In our study, VDL molecular collisions could change the stochastic component so that their Brownian motion behaviors are also changed. On the other hand, there is no pure random system in the nature. They are still following normal distribution in a large sample.

- Motility measurements may be signigficantly influenced by the number of cells (both leukemia and stromal cells) in each well. Have the authors looked at this?

Response: This is a great point. Stromal cells are seeded  at  beginning of the experiment equally in the  plates,  however we  did not account for  eventual  cell death at Day 7.

- What does Figure 2e refers to? Viability with or without stromal support?

Response: Figure 2e showed the cell viability after 7 days culture with stromal support.

- Was the cell viability of LAX56 and ICN24 checked? Also, I would expect much higher viability of ICN24 cells as these are from a relapse sample. Have the authors checked this?

Response:  The viability was checked. Please see Figure 4 for your reference. Figure 4a, 4d, and 4g showed the viability of LAX7R, LAX56 and ICN24. ICN24 was the diagnosis sample and LAX56 is the relapsed sample (Table 1). Unexpectedly, viability of VDL treated ICN24 group was not lower than that of LAX56. The strength of this paper is that patient-derived B-ALL cells are used. One explanation is, that the growth and viability if patient-derived cells are dependent on stroma and viability may vary. Therefore, the comparison between Media and VDL of the same B-ALL case is useful.

- From the images in Figure 2 it seems that there are more chemoresistant LAX7R cells without stromal support, which is counterintuitive. Please explain.

Response: Yes, we agree reviewer that it should be more chemoresistant LAX7R cells with stromal support. Please see the figure 1 for the illustration of 8-well chamber (right-bottom corner). In the study, the time-lapse microscopy is automatically switching each well to monitor cells. The view of the microscopy is much smaller than the well size. In the experiments, it is hard to make sure the cells distribute equally. Therefore, it is possible to observe more chemoresistant LAX7R cells without stromal support within the specific view of time-lapse microscopy. However, it is more chemoresistant LAX7R cells with stromal support than without stromal support by using whole field microscopy.

- The discussion of figure 4 is very confusing.

Response: The discussion is revised ( 311-318).

- What was the rationale to test integrin blockage? Was the experiment in figure 4 done on stromal support? Do stromal cells express the integrin ligands?

Response:  Integrins play a crucial role in linking the extracellular and intracellular skeletons. Cytoplasmic linker molecules are thought to mediate integrin-actin binding, and likely act as potential dynamic mediators of cell anchorage (Blystone SD. Integrating an integrin: a direct route to actin. Biochim Biophys Acta. 2004;1692(2-3):47-54. doi: 10.1016/j.bbamcr.2004.04.011. PubMed PMID: 15246678). Given that the organization of actin filament networks is critical for cell motility we hypothesized that integrin-actin-binding plays a role in motility and deformability of leukemia cells.

- I would expect if integrins are relevant for the adhesion of B-ALL cells that their inhibition resulted in a significant increase in migration, as shown for the abscense of stromal support.

Response: As the

- Previous data that are very relevant for the type of biological phenomena the authors are studying were overlooked (check the papers: PMID 26058075, 27750279, 30262563). I suggest that the authors perform an extensive literature review on the topic and redefine their references.

Response: References have been included and updated.

Line 52~59 for reference 16;

Line 276~288 for reference 22

Line 289~291for reference 23

We also added (PMID: 30422351).: “Of note, it was also shown that using intravital imaging, AML cells are migratory, and in contrast with T‐ALL, chemoresistant AML cells become less motile (PMID: 30422351). Further  studies are  warranted to distinguish motility  differences in  different  subtypes of leukemia  and  different drug treatments to elucidate  further the  effects on  motiliy”

Round 2

Reviewer 1 Report

As the authors’ replay, it needs to provide the data indicating the drug-resistance in the drug-treated cells. It is understandable for the difficulty of COVID-19 pandemics. Thus, the following suggestions may be reasonable for the final acceptance of the manuscript.

  • The manuscript may focus on exploring the motility of the leukemia cells by Time-lapse image, not emphasize the drug-resistance. If so, it is better to change the title as ” Characterizing the motility of chemotherapeutics-treated Acute lymphoblastic leukemia cells by time-lapse imaging” or avoid using the “chemotherapeutic resistant”.
  • Also, change the statements concerning the “chemotherapeutic resistant” and/or drug resistance in the abstract, introduction, methods, and results section. In the discussion, you may explain the possibility of the drug-resistance in the drug-treated cells.

Hopefully, these suggestions are valuable for the final acceptance of the manuscript.

Author Response

Reviewer #1

Comments and Suggestions for Authors

As the authors’ replay, it needs to provide the data indicating the drug-resistance in the drug-treated cells. It is understandable for the difficulty of COVID-19 pandemics. Thus, the following suggestions may be reasonable for the final acceptance of the manuscript.

Response: We thank the reviewer for her/his comments and review.

The manuscript may focus on exploring the motility of the leukemia cells by Time-lapse image, not emphasize the drug-resistance. If so, it is better to change the title as "Characterizing the motility of chemotherapeutics-treated Acute lymphoblastic leukemia cells by time-lapse imaging" or avoid using the "chemotherapeutic resistant".

Response: The title is revised as the reviewer suggested: "Characterizing the motility of chemotherapeutics-treated acute lymphoblastic leukemia cells by time-lapse imaging". Thank you.

Also, change the statements concerning the "chemotherapeutic resistant" and/or drug resistance in the abstract, introduction, methods, and results section. In the discussion, you may explain the possibility of the drug-resistance in the drug-treated cells.

Response: Statements concerning the "chemotherapeutic resistant" and/or drug resistance in the abstract, introduction, methods, and results section are revised as suggested. The possibility of the drug-resistance in the drug-treated cells in the Discussion section is explained now as suggested (Line 301~304 and 313~314). Thank you.

Hopefully, these suggestions are valuable for the final acceptance of the manuscript.

Response: We thank the reviewer for her/his valuable suggestions.